# Neglected risks in the Chinese insurers' misconduct caused by digitalization

**Jiandi Zhang[1], Xiaoqing Guo [2]\*, Rui Xu[3], Zhengfa Yang[4]**

**1** Shanghai Lixin University of Accounting and Finance, Shanghai, China, **2** School of Insurance, Central University of Finance and Economics, Beijing, China, **3** Guangdong Branch, Postal Savings Bank of China, Guangzhou City, Guangdong Province, China, **4** China North Industries Institute of Planning and Research, Beijing, China

\* 2022110077@email.cufe.edu.cn

## Abstract

Digital transformation offers opportunities for innovative growth while also introducing emerging risks of misconduct in the insurance industry. This study examines the impact of digital transformation on insurers' misconduct, focusing on 72 Chinese insurers from 2010 to 2021. Through text analysis, we quantify insurers' digital transformation and misconduct. The results indicate that digital transformation tends to increase misconduct, particularly among property insurers and those with joint or foreign ownership. The issuance of digital policy and the adoption of technologies that exceed managerial capacity exacerbate misconduct. Digital transformation drives misconduct through the channels of market expansion and financial pressure. Furthermore, insurers with larger market shares, lower commissions, and operational costs are more prone to misconduct during their digital transformation. Conversely, higher insurance density, market penetration, premium income growth, and lower economic uncertainty help mitigate misconduct.

## 1. Introduction

In the global insurance industry, the rapid embrace of digital technologies has significantly reshaped key operational areas, including product marketing, intelligent underwriting, automated claims processing, and fraud detection (Gatteschi et al., 2018 [1]; de Andrés & Gené, 2024 [2]). While the benefits of technological progress and digitalization are well-documented (Ding et al., 2024 [3]; Zhang et al., 2024 [4]), their early adoption phase may also open new channels for misconduct and regulatory violations.

Corporate misconduct constitutes a unique operational risk, characterized by unethical business practices that break laws or harm stakeholders (Xia et al., 2023 [5]). Alarmingly, such misconducts exacerbate long-term inefficiencies, which in turn result in regulatory penalties, reputational damage, and market instability (Makau et

**Data availability statement:** All data files are available from the figshare database. DOI: https://doi.org/10.6084/m9.figshare.30312571.v1.

**Funding:** The author(s) received no specific funding for this work.

**Competing interests:** The authors have declared that no competing interests exist.

al., 2021 [6]). The Chinese insurance market provides a representative context for analysis, as its regulatory framework is characterized by uniform rules and comprehensive information disclosure, offering consistent violation records and standardized penalty criteria. Consistently, regulatory data from China show an increase in insurers' misconduct over the past decade—a trend that aligns with the rapid digital transformation of the insurance industry. In fact, Since the State Council launched the "Broadband China" plan in 2013, digital transformation in China has accelerated. Concurrently, data from the China Financial Regulatory Administration show that instances of insurers' misconduct have increased alongside advancements in digital technology. Specifically, there has been a rapid increase in penalties for misconduct since 2017. In 2024, Chinese insurers received 2,762 misconduct penalties, with total fines amounting to 280 million yuan, representing an increase of 137.3% compared with 2017.

This paradox between rapid digitalization and rising misconduct prompts critical questions about the unintended consequences of insurers' digital transformation. Therefore, the following questions are raised: Are there risks of misconduct in the digital transformation of insurers? Why does the digital transformation of insurers present new challenges in managing misconduct while advancing the industry? Answering these questions offers valuable insights for regulators in emerging markets seeking to balance innovation and compliance.

However, existing literature has predominantly emphasized the positive impacts of digitization on market value (Fritzsch et al., 2021 [7]), sales innovation (Eckert et al., 2021 [8]), product customization (Radwan, 2019 [9]), pricing precision (Radwan, 2019 [9]), customer service optimization (e Sá et al., 2024 [10]), and fraud reduction (Radwan, 2019 [9]), with limited attention to potential risks, especially misconduct ones. Consequently, understanding whether digital transformation in the insurance sector is associated with an increase in misconduct contributes to corporate governance and regulatory oversight, especially in light of the growing prevalence of digital technologies among insurers.

Using data from Chinese insurers between 2010 and 2021, this empirical study is the first to provide evidence on how digital transformation influences insurers' misconduct and the underlying mechanisms in an evolving insurance market. We compile annual disclosure reports data from 72 insurers in China (the majority of which are non-listed) and construct a dictionary for digital transformation through text analysis. Next, following Raghunandan (2024) [11], we extract penalty amounts and fines imposed on insurers from the violation announcements issued by the regulator. Based on the construction of the core variables, we introduce additional control variables and employ firm and year fixed effects to conduct an empirical analysis of the impact of digital transformation on insurers' misconduct. The results suggest that digital transformation increases insurers' misconduct, and robustness as well as endogeneity tests confirm the validity of these findings.

However, it is worth noting that there is a potential controversy in using government penalty data to construct the indicator of insurers' misconduct: Does the rise in government-imposed penalties reflect an actual increase in corporate misconduct,

or is it merely a result of heightened regulatory scrutiny and an increased likelihood of detection during digitalization? To validate the appropriateness of our variable selection, we employ a range of methods, including the incorporation of interaction terms, the substitution of dependent variables, and quasi-natural experiments. The empirical analysis demonstrates that digital transformation indeed exacerbates insurers' propensity to engage in misconduct.

We then conduct a heterogeneity analysis by insurer type, capital structure, the implementation of the "Broadband China" policy, and the digital transformation category. The results show that property insurers and those with joint or foreign ownership are more affected by digital transformation regarding misconduct. The "Broadband China" policy accelerates insurers' digital transformation but also exacerbates misconduct. While basic technological advancements in digital transformation help reduce misconduct, more advanced technology development and application have the opposite effect.

Furthermore, we construct a channel regression model, employing premium income and the cost ratio as proxies for market expansion and financial pressure variables, respectively. The result shows that the digital transformation of insurers increases misconduct by increasing premium income and cost ratio. In addition, we identify moderate variables to explore complementary tests as as evidence of plausible channels. Internally, firms with greater market power face stronger incentives for misconduct, whereas lower handling fees and operating expenses constrain opportunistic behavior. Externally, mature insurance markets curb misconduct, while economic uncertainty exacerbates it. However, changes in regulatory intensity do not significantly mitigate misconduct, suggesting that regulatory improvements alone cannot fully explain the rise in penalties.

Our research makes three key contributions:

First, we construct measurement indicators for digital transformation and misconduct through test analysis for insurers. In developing the digital transformation dictionary, we not only refer to previous literature but also incorporate terms that specifically reflect insurers' digital operations, resulting in a comprehensive and highly relevant dictionary. For the misconduct measurement indicator, we utilize violation penalty data to address the issue of incomplete information from some insurers and to standardize the measurement process. Additionally, unlike most studies that only focus on listed insurers, our research adopts a broader perspective by manually collecting data from both listed and non-listed insurers. This expanded dataset mitigates sample bias and strengthens the reliability of research on digital transformation measurement and its associated risks.

Second, we highlight the often-overlooked risks of digital transformation. Existing studies primarily focus on the economic benefits of digital transformation, but overlook its potential risks, especially misconduct risks arising from business model reconstruction and operational innovation. The majority of extant literature emphasizes the positive impacts of digital transformation on corporate efficiency. However, our study is the first to systematically disclose the negative effect that digital transformation can lead to an increase in misconduct during its development.

Third, we investigate the mechanism underlying the impact of digital transformation. The insurance industry's heavy reliance on data and risk management introduces unique complexities in analyzing how digital transformation affects insurers' misconduct, resulting in a dearth of prior research in this area. By incorporating premium income and the claims ratio into the channel model, we provide novel evidence that the market expansion and financial pressures resulting from digital transformation can lead to an increase in misconduct. In addition, by considering both insurers' internal operations and external environments, we select relevant indicators to construct cross-multiplication terms and successively add the benchmark regression model. This approach enables us to analyze the moderating effects of different factors on the impact of digital transformation on insurers' misconduct and realize the complementary tests of plausible channels. This mechanism analysis enriches the existing theoretical framework and offers a novel perspective on the multifaceted roles of digital transformation.

The remainder of this study is organized as follows. Section 2 reviews the relevant literature and formulates the hypotheses. Section 3 provides an overview of the data sources, variable selection criteria, and benchmark model specifications. Section 4 presents the baseline regression results, along with robustness checks and endogeneity tests. It also examines

the heterogeneous characteristics of digital transformation that influence insurers' misconduct. Section 5 explores the underlying mechanisms through which digital transformation influences insurers' misconduct by channel analysis and complementary tests. Lastly, the final section briefly concludes.

## 2. Literature review and hypotheses development

Enterprise misconduct has long been a significant barrier to the sustainability of the insurance industry and the sound functioning of insurers (Schiro, 2006 [12]). Scholars have identified a strong connection between insurers' internal management and misconduct (Rizwan, 2019 [13]; Ben, 2024 [14]; Gunaseelan et al., 2024 [15]). Some argue that changes in the external regulatory environment impact insurers' misconduct (Chen & Hieber, 2016 [16]; Srbinoski et al., 2022 [17]; Koziol & Kuhn, 2023 [18]). Others have explored the consequences of misconduct for insurers and the broader industry (Makau et al., 2021 [6]; Talesh & Filho, 2023 [19]). Despite extensive research on company management, industry supervision, and economic outcomes, there is a notable gap in the literature regarding the impact of digital transformation on insurers' misconduct, especially as technologies such as big data, cloud computing, Internet of Things (IoT), blockchain, and artificial intelligence (AI) continue to evolve and integrate into the sector.

To the best of our knowledge, only Wang & Han (2023) [20] and Wang et al. (2024) [21] empirically demonstrate that digital transformation reduces corporate fraud and illegal corporate acts. In the field of insurance, scholars have not discussed this topic in depth. Nevertheless, emerging technologies such as blockchain, cloud computing, and big data enable the automation of insurance transaction processes. On the one hand, this reduces misconduct by insurers, such as misleading sales, by minimizing manual intervention. On the other hand, it also significantly lowers the labor costs associated with insurers' operations. According to resource allocation theory (Maritan & Lee, 2017 [22]), this allows insurers to allocate more resources to misconduct management, thereby reducing misconduct risks. Furthermore, the consensus-based features of digital technologies enhances information transparency in the insurance market, making it easier for regulatory authorities to identify and address insurers' misconduct. In line with deterrence theory (Scholz, 1997 [23]), this helps prevent insurers from engaging in improper practices.

Although there are no empirical findings confirming that digital transformation reduces insurers' misconduct, the aforementioned analysis indicates that digital transformation may contribute to a reduction in such misconduct through multiple channels, including optimizing insurers' internal management and external environments. Meanwhile, there is no doubt that the research findings of many scholars support the positive impact of the application of digital technology on the management of insurers and the development of the insurance market (Fritzsch et al., 2021 [7]; Eckert et al., 2021 [8]; Radwan, 2019 [9]; e Sá et al., 2024 [10]). Therefore, we propose the following hypothesis.

H0: Digital transformation decreases insurers' misconduct.

However, H0 represents the outcome that most scholars expect to see, and it is also a viewpoint supported by related studies in other industries. Nonetheless, it is worth noting that in the insurance industry, the above hypothesis derived from such analyses is partially inconsistent with the actual data (As detailed in footnote ①). Moreover, a few theoretical studies suggest that digitalization introduces novel risks for financial institutions, complicating misconduct management (Jagtiani & John, 2018 [24]).

Insurers have made significant progress in product R&D, channel innovation, and regional expansion due to digital transformation. However, insurer management is distinctive in its core operations, relying on the handling of sensitive data and complex risk assessments within a highly regulated environment. From the internal perspective, the substantial resources and investment needed for technological innovation might leave insurers with insufficient means to boost productivity and compliance risk management efficiency, thus facing the "innovator's dilemma" (Christensen, 2015 [25]). Studies have indicated that digital transformation can reshape insurers' business models (Ostrowska, 2021 [26]; Srivastava et al., 2024 [27]), with these changes potentially resulting in misconduct risks (Ben, 2024 [14]). From a regulatory environment perspective, the preliminary implementation of emerging technologies often leads to an uncertain legal and

 

regulatory environment (a regulatory vacuum) (Jagtiani & John, 2018 [24]), which can give rise to issues such as privacy leaks, technology misuse, and market unfairness. According to information asymmetry theory (Arrow, 1963 [28]), the complexity of digital technology may exacerbate information asymmetry among firms, users, and regulators (Calderon-Monge & Ribeiro-Soriano, 2024 [29]), thereby triggering and worsening misconduct.

Accordingly, we further propose the following hypothesis based on the current state of the insurance industry:

H1: Digital transformation increases insurers' misconduct.

## 3. Research design

### 3.1. Data sources

Using data from 72 Chinese insurers spanning 2010–2021, we collect information from three primary sources: penalty announcements issued by the National Administration of Financial Regulation, insurers' annual disclosure reports, and the China Insurance Yearbook. We choose this period because financial data on insurance companies has been relatively complete since 2010, and the most recent available annual report data is from 2021. To ensure sample homogeneity, we exclude asset management, brokerage, and reinsurance firms. As of the end of 2021, there are 235 insurers in China. Due to missing or abnormal values, we exclude insurers established after 2010 and 88 insurers with incomplete disclosure. After addressing variable matching concerns, we retain a final sample of 72 insurers for analysis. To enhance the credibility of the conclusions, we employ a variety of methods, including linear interpolation, extreme value elimination, standardization, and logarithmic transformation for missing and extreme data.

### 3.2. Variable selection

**3.2.1. Insurers' digital transformation.** The core explanatory variable of our study is the degree of digital transformation of insurers, measured from their annual information disclosure reports. Existing literature employs various topic modeling approaches, such as latent Dirichlet allocation (LDA) and non-negative matrix factorization (NMF) (Fritzsch et al., 2021 [7]; Wang and Zhang, 2023 [30]), while some recent studies utilize large language models (LLMs) to construct digital transformation indicators (Jin et al., 2024 [31]). These methods do not require predefined keywords and can capture latent topics, but they also suffer from limitations such as low interpretability and high requirements for text standardization. Consequently, they are typically applied to studies covering entire industries rather than focusing on a specific sector. In light of these considerations, we construct our digital transformation indicator using a Python-based dictionary approach, which offers greater interpretability (Wu et al., 2021 [32]).

First, we construct a digital transformation dictionary by integrating vocabularies from both established research and insurance industry-specific sources. Following the approach of Zhen et al. (2023) [33], we categorize the dictionary entries into two groups: Basic technologies and technology development & applications.

Second, we manually collect information disclosure documents of 72 Chinese insurers, including both listed and non-listed firms. Globally, comprehensive firm-level data on insurers' digital transformation remains scarce, particularly for non-listed insurers due to the lack of annual disclosure reports. Consequently, most existing studies focus primarily on listed insurers with accessible reports, leading to sample bias and limited reliability of findings. To address this issue and enhance research reliability, we adopt a broader perspective by manually collecting data from both listed and non-listed insurers. Each report is converted into a TXT file for text analysis, with file names formatted as 'insurer-year' to establish a basic corpus.

Third, we preprocess the corpus, which includes applying Jieba for Chinese word segmentation, removing stop words across all texts, and excluding negative sentences containing digital transformation vocabulary.

Fourth, to ensure that negative statements such as "we are not currently considering using digital risk control technology" are not misidentified, we construct a negative word dictionary. Specifically, if a negative word appears in the same sentence as a thesaurus term, that term is excluded from the word frequency statistics.

Finally, leveraging the constructed dictionary and corpus, we conduct text extraction and word frequency analysis to quantify digital transformation. These frequencies are matched with the corresponding companies and years, producing annual firm-level word frequency data on digital transformation (*digital_count* ). Since this data is right-skewed, we apply a logarithm transform to form the basic measurement -- the annual digital transformation degree of insurers (*digit*1). Table A in the S1 File document reports the word categories, their frequencies, and corresponding percentages of the total sample.

**3.2.2. Insurers' misconduct.** The dependent variable in our study is insurers' misconduct. Following Raghunandan (2024) [11], we construct a measurement indicator for insurers' misconduct using government penalty data, which is sourced from publicly available information disclosed by the National Financial Regulatory Administration (NFRA). First, we employ machine learning-based web scraping techniques to extract texts on insurers' misconduct penalties from the official information of the regulatory department. After organizing the texts, we apply extraction, matching, and segmentation techniques in text analysis to obtain details like company names, penalty amounts, events, and corresponding city and time information. We also match the penalty records of insurance company branches with parent companies. Then, at the company-year level, we construct an annual penalty amount indicator of insurers (*sum_fine* ) to quantify their misconduct. Notably, for insurers with no penalties in a year, we set their annual penalty amounts and penalty times to zero to prevent data missing.

In the context of digital transformation, government penalty data serves as the foundation for constructing enterprise misconduct indicators. This raises a pertinent question: Does the surge in government penalties with an escalation in corporate misconduct coincide, or is it merely a reflection of heightened government oversight? Indeed, the digital transformation driven by enterprises themselves will render business processes and internal risk management systems more transparent, thereby facilitating the detection of misconduct by regulators. Meanwhile, advancements in digital technology have the potential to enhance regulatory tools at the national level, thereby facilitating the identification of misconduct by insurers. Therefore, it is imperative to establish effective methodologies for distinguishing between this heightened level of regulation and a genuine escalation in corporate misconduct, which increases the number and severity of government penalties. This is necessary to more precisely ascertain the impact of digital transformation on violations, and this issue is a focal point of our research.

In the subsequent empirical analysis, we address the aforementioned questions by constructing interaction terms, replacing dependent variables, and employing quasi-natural experiments. We conclude that digital transformation has played a direct role in changing internal risk management and compliance mechanisms, leading to more actual misconduct, and not just because regulators have improved their oversight as a result of digital transformation. The empirical results obtained in this study substantiate the validity of the variable construction, as will be explained further below.

**3.2.3. Other variables.** We also use company size (*Size*), number of employees (*NMS*), return on net assets (*ROA*), growth rate of owners' equity (*EGR*), and net asset ratio (*NAR*) as control variables. Variable symbols, construction methods, data sources, and references of the main regression model are summarized in Table 1.

**Table 1. Variable Selection.**

| Symbols | Construction Method | Data Source | Reference |
|---|---|---|---|
| *sum_fine* | *Fine of Regulatory Penalty* | NFRA | Ming et al. (2023) [34] |
| *digit*1 | ln(*digital_count*) | The annual information disclosure reports of insurers | Wu et al. (2021) [32] |
| *Size* | ln(*Asset*) | China Insurance Yearbook | Morara and Sibindi. (2021) [35] |
| *NMS* | *Number of marketing staff* | | Anderson et al. (2018) [36] |
| *ROA* | *Net Income ÷ Average Total Asset* | | Morara and Sibindi (2021) [35] |
| *EGR* | *ΔOwners' Equity ÷ Owners' Equity* | | |
| *NAR* | *Net Assets ÷ Total Assets* | | |

**3.2.4. Descriptive analysis and correlation analysis of variables.** The descriptive statistical results and the correlation analysis results of the model variables are presented in Table 2 and Table 3, respectively. The results in Table 2 show that the means (standard deviations) of *sum_fine* and *digit*1 are 1.55 (5.20) and 0.38 (0.73), respectively, which means that there are relatively large differences in the misconduct behavior and degree of digital transformation of different insurers. Therefore, it is necessary to clarify the influence mechanism and channels through which digital transformation affects the insurers' misconduct. The data analysis of control variables is generally reasonable.

The results in Table 3 show that there is a significant positive correlation between *sum_fine* and *digit*1. It reflects that the progress of insurers' digital transformation may bring entirely new risks and challenges to the company's operations, intensifying their misconduct behavior. However, further testing and exploration are required to ascertain the relevant trend and mechanism.

## 3.3. Model construction

To examine the direct impact of digital transformation on insurers' misconduct, we construct the following model for the baseline regression analysis:

$$sum\_fine_{i,t} = \alpha_0 + \alpha_1 digit1_{i,t} + A \times controls_{i,t} + \mu_i + \gamma_t + \xi_{i,t} \qquad (1)$$

Where $i$ represents insurer, $t$ represents year, $sum\_fine_{i,t}$ represents the dependent variable calculated based on penalty amount for misconduct – insurers' misconduct, $digit1_{i,t}$ represents the core explanatory variable calculated based on word frequency of digital transformation – the degree of digital transformation of insurers, $controls_{i,t}$ represents control variables, including company size ($Size$), number of employees ($NMS$), return on net assets ($ROA$), growth rate of owners' equity ($EGR$), and net asset ratio ($NAR$). The control variables used in different regression models are adjusted according to the actual research needs, and the specific indicators are measured in Table 1. $\mu_i$ and $\gamma_i$ represent the company fixed effect and the year fixed effect, respectively. $\xi_i$ represents the random disturbance term.

**Table 2. The Descriptive Statistical Results.**

| Variable | N | Mean | Standard Deviation | Minimum value | Maximum value |
|---|---|---|---|---|---|
| *sum_fine* | 669 | 1.55 | 5.20 | 0 | 69.14 |
| *digit*1 | 669 | 0.38 | 0.73 | 0 | 5.39 |
| *Size* | 618 | 9.71 | 1.91 | 5.78 | 15.69 |
| *NMS* | 590 | 8.53 | 1.99 | 4.20 | 14.27 |
| *ROA* | 623 | −0.45 | 29.88 | −345.05 | 218.92 |
| *EGR* | 634 | 8.78 | 1.00 | −1.12 | 15.44 |
| *NAR* | 641 | 32.50 | 166.37 | −0.53 | 3237.85 |

**Table 3. The Correlation Analysis Results.**

| VarName | 1 *sum_fine* | 2 *digit*1 | 3 *Size* | 4 *NMS* | 5 *ROA* | 6 *EGR* | 7 *NAR* |
|---|---|---|---|---|---|---|---|
| 1 | 1 | | | | | | |
| 2 | 0.19*** | 1 | | | | | |
| 3 | 0.38*** | 0.34*** | 1 | | | | |
| 4 | 0.43*** | 0.26*** | 0.84*** | 1 | | | |
| 5 | 0.12*** | 0.13*** | 0.33*** | 0.25*** | 1 | | |
| 6 | 0.23*** | 0.36*** | 0.47*** | 0.39*** | 0.20*** | 1 | |
| 7 | −0.02 | −0.03* | −0.07* | −0.04 | 0.02 | −0.03 | 1 |

## 4. Empirical analysis

### 4.1. Benchmark regression analysis and Robust test

Using Equation (1) as a baseline, we first perform an ordinary least squares (OLS) regression without incorporating two-way fixed effects or control variables (Table 4, Column (1)). The results suggest that advancements in digital transformation significantly increase insurers' misconduct. This finding may be attributed to two key factors. First, the features of digital technologies, such as interconnection and sharing, have a significant impact on enhancing institutional information transparency (Yáñez and Guerrero, 2023 [37]), easily exposing insurers' illegal operations to regulators. Second, the emerging digital technologies are immature, and industrial application introduces new challenges to insurers' development (Ciborra, 2006 [38]; Gatzert and Schubert, 2022 [39]).

One potential concern is that regulatory penalties may not be randomly assigned. Financial regulators may systematically target firms that are larger, riskier, or located in certain regions. In addition, the incomplete selection of relevant control variables may lead to bias in the research results. To account for systemic regulatory differences and potential omitted variable bias, we sequentially introduce two-way fixed effects and control variables in our panel data regression analysis. The regression results with two-way fixed effects (Table 4, Column (2)), control variables (Table 4, Column (3)), and both (Table 4, Column (4)) reconfirm the previous findings. Notably, the panel two-way fixed effects model with control variables has a higher $R^2$ value. Therefore, the following research would mainly rely on this model with a better explanation.

Based on the benchmark regression analysis, we conduct robustness tests using indicator substitution, data winsorizing, and model variations. Firstly, to evaluate the effect of indicator construction on our results, we replace the core explanatory variable (Table 4, Column (6)) and the dependent variable (Table 4, Column (9)) and re-run the regression analyses on the panel data. The robustness tests for the core explanatory and dependent variables adhere to the

**Table 4. Benchmark Regression Analysis and Robust Test.**

| Dependent Variable | | | | | |
|---|---|---|---|---|---|
| **Explanatory Variable** | **sum_fine** | | | | |
| | (1) | (2) | (3) | (4) | (5) |
| *digit*1 | 1.32*** | 0.81*** | 0.72** | 0.58* | 0.31*** |
| | (4.85) | (2.77) | (2.32) | (1.76) | (3.27) |
| Control Variable | No | No | Yes | Yes | Yes |
| Company fixed effect | No | Yes | No | Yes | Yes |
| Year fixed effect | No | Yes | No | Yes | Yes |
| N | 669 | 669 | 557 | 557 | 557 |
| $R^2$ | 0.03 | 0.12 | 0.20 | 0.13 | 0.22 |

| Dependent Variable | | | | | |
|---|---|---|---|---|---|
| **Explanatory Variable** | **sum_fine** | | | **sum_i_id** | **sum_fine'** |
| | (6) | (7) | (8) | (9) | (10) |
| *digit*1 | | 0.58* | 0.19*** | 2.37* | 0.44** |
| | | (1.76) | (3.03) | (1.80) | (3.08) |
| *digit*2 | 0.34** | | | | |
| | (2.10) | | | | |
| Control Variable | Yes | Yes | Yes | Yes | Yes |
| Company fixed effect | Yes | Yes | Yes | Yes | No |
| Year fixed effect | Yes | Yes | Yes | Yes | No |
| N | 556 | 557 | 557 | 533 | 618 |
| $R^2$ | 0.15 | 0.12 | —— | 0.17 | 0.11 |

methods used in the baseline regression, drawing on Wu et al. (2021) [32] and Ming et al. (2023) [34]. We utilize text analysis to extract data on the number of penalties from the NFRA (*digital_count*).

To address potential greenwashing in annual reports, whereby insurers may exaggerate or even fabricate their digital transformation efforts, we train and fine-tune a large language model (LLM) to further filter digital terms. The model is designed to accurately distinguish between action-level and non-action-level statements. For example, a sentence containing a digital transformation term is retained only if it explicitly describes a concrete action undertaken by the firm to advance digital transformation. This approach preserves the applicability of text-based measures while mitigating bias arising from greenwashing, thereby providing a more accurate reflection of firms' actual progress in digital transformation. Our objective is to minimize the impact of greenwashing, albeit at the risk of underestimating the extent of digital transformation.

The construction proceeds as follows. First, we label all sentences containing digital transformation terms and identify the corresponding firm names and report years. Second, we use a bootstrap sampling method to select 1,000 sentence pairs for manual annotation, and employ a custom-designed prompt to train the LLM. Third, after training, we classify each sentence in the sample according to its semantic level and retain only keywords identified as "action-level" to construct the final digital transformation measure (*digit2*). The LLM is fine-tuned based on Qwen2.5, achieving over 95% classification accuracy on the validation set.

In robustness tests, we use the LLM-based digital transformation measure (*digit2*) and the number of regulatory penalties (*digital_count*) as alternative independent and dependent variables, respectively. The results show that the digital transformation coefficient remains significantly positive, confirming the conclusions of our baseline regression.

It is worth noting that, using both the number of penalties and the total penalty amount as dependent variables, the regression analysis indicates that digital transformation has a significantly positive effect on insurers' misconduct. Major misconduct is more likely to be detected by regulators, and a key feature of such misconduct is the imposition of significant financial penalties. If the increase were driven solely by enhanced information transparency due to digitalization, we would expect to see a rise primarily in minor misconducts, with little change in total penalty amounts. However, if misconduct itself has genuinely increased, then both the number and the amount of penalties would rise significantly. The observed significant increase in both metrics suggests that not only minor but also major misconduct is on the rise, supporting the conclusion that digital transformation is directly contributing to the increase in insurers' misconduct.

Secondly, to mitigate the impact of extreme values, we conduct robustness tests by winsorizing data. We respectively replace outlier values above the 95th (Table 4, Column (5)) and below the 5th percentile (Table 4, Column (7)) with their respective thresholds and re-run the regression. The results confirm that our benchmark research conclusions hold.

Finally, to address the impact of censored data on research conclusions, we employ the Tobit model, which effectively handles truncated data, to perform a regression analysis on the panel datasets. Besides, we also construct a binary variable *sum_fine′* based on the value of the dependent variable *sum_fine*. When *sum_fine* > 0, *sum_fine′* takes the value of 1; otherwise, it takes the value of 0. Then we conduct a logit model regression analysis with *sum_fine′* as the dependent variable (Table 4, Column (10)). The regression results further confirm the validity of our benchmark regression findings. Overall, the baseline findings exhibit strong robustness.

We also examine the possibility of a U-shaped or inverted U-shaped nonlinear relationship in the model. We include the quadratic term of the independent variable and conduct both the U-test and the White test. While the coefficient of the independent variable remains statistically significant, the model does not pass the nonlinearity tests. Therefore, we conclude that there is no evidence of a nonlinear relationship between digital transformation and insurers' misconduct.

### 4.2. Endogeneity test

This study faces notable endogeneity concerns that may introduce significant bias or distortion into the empirical results. First, reverse causality manifests primarily in the interaction between digital transformation and misconduct. Internal

misconduct within firms is an issue that both managers and regulators seek to resolve (Rizwan, 2019 [13]; Ben, 2024 [14]; Gunaseelan et al., 2024 [15]). When misconduct increases, firms may be compelled to implement internal governance reforms, such as enhancing process transparency and standardizing operations, which in turn drive digital transformation. Secondly, omitted variable bias may arise from unobserved factors that simultaneously influence both digital transformation and misconduct. Digital transformation could increase the exposure to misconduct risk, making it easier for regulators to detect non-compliance. However, the key variable of regulation is often difficult to measure directly, leading to omitted variable bias.

To address endogeneity concerns and enhance the accuracy and reliability of our findings, we employ three methodological approaches: A multi-period difference-in-differences (DID) model, an instrumental variable (IV) test, and the propensity score matching (PSM) method.

**4.2.1. Multi-period DID model.** We implement a multi-period DID regression analysis to estimate its dynamic effect on insurers' misconduct while controlling for firm-specific and time-varying factors. In our design, insurers that undergo digital transformation serve as the treatment group, while those that never adopt digital transformation serve as the control group. The model is shown in Equation (2).

$$sum\_fine_{i,t} = \alpha_0 + \alpha_1 du_{i,t} \times dt_{i,t} + A \times controls_{i,t} + \mu_i + \gamma_t + \xi_{i,t} \qquad (2)$$

Where $du_{i,t}$ is an individual dummy variable, during the sample period, the variable is 1 for insurers with digital transformation and 0 otherwise. $dt_{i,t}$ is a time dummy variable that equals 1 in the year an insurer initiates digital transformation and in all subsequent years, and 0 otherwise. All other symbols are defined as in the preceding sections.

Based on Equation (2), we conduct multi-period DID regression analyses at the company and year levels, both without (Table 5, Column (1)) and with (Table 5, Column (2)) control variables. Through two rounds of difference processing on the digital transformation indicators of insurers in the experimental and control groups, we can effectively eliminate individual inherent differences and biases from time trends unrelated to the experimental group, and obtain the "net effect" of insurers' digital transformation on misconduct.

In addition, since the dummy variable may neglect the degree of digital transformation, we also examine the regression when the annual digital transformation word frequency of insurers (*digital_count*), regarded as a quantitative indicator of transformation intensity, is added to the interaction term $du \times dt$ (Table 5, Column (3)). The coefficient of $du \times dt \times digital\_count$ represents the impact of the intensity of insurers' digital transformation on their misconduct. The triple interaction term identifies whether the increase in misconduct is driven by firms' endogenous behavioral changes rather than merely improved regulatory detection. If penalties increase due to more misconduct resulting from digital transformation, the treatment group would show a significant rise in misconduct, irrespective of the regulatory environment. Conversely, if the increase is solely due to higher detection probabilities, all digitized firms would experience a similar effect rather than a non-uniform pattern. To test the reliability of multi-period DID, we also conduct a parallel trend test. The results are shown in Column (4) in Table 5. $d\_i$ stands for the $i$ year before the policy shock, and $di$ stands for the $i$ year after it.

Table 5 shows that the coefficients of $du \times dt$ in Column (1) and (2) and $du \times dt \times digital\_count$ in Column (3) are positive, aligning with the benchmark regression results. The results further exclude alternative reasons for increased misconduct detection rates. Notably, the coefficient of the latter is much smaller, indicating that when the digital transformation degree is higher, the negative influence of digital transformation on their misconduct is mitigated. This could be attributed to the fact that a more advanced stage of digital transformation indicates the implementation of more mature technological applications and improved compliance management, which consequently reduces the overall impact. In addition, in Column (4) of Table 5, the coefficients of the indicators after the policy implementation year are all significantly negative at the 10% level. In contrast, those prior to implementation are insignificant. This validates the parallel trend test hypothesis.

**Table 5. Endogeneity Test.**

| Dependent Variable | | | | | | |
|---|---|---|---|---|---|---|
| Explanatory Variable | Multi-Period DID Analysis | | | | Two-Stage Least Squares | |
| | sum_fine | | | | digital_count | sum_fine |
| | (1) | (2) | (3) | (4) | (5) | (6) |
| $du \times dt$ | 1.37*** (3.27) | 1.18*** (2.74) | | | | |
| $du \times dt \times digital\_count$ | | | 0.16** (2.53) | | | |
| instrumentvariable | | | | | 1.53*** (2.48) | |
| digital_count | | | | | | 2.06*** (4.88) |
| $du \times d\_3$ | | | | -1.10 (−1.21) | | |
| $du \times d\_2$ | | | | -0.28 (−0.90) | | |
| $du \times d\_1$ | | | | -0.94 (−1.51) | | |
| $du \times current$ | | | | -1.61* (−2.01) | | |
| $du \times d1$ | | | | -1.22* (−1.81) | | |
| $du \times d2$ | | | | -0.98* (−1.68) | | |
| $du \times d3$ | | | | -1.38* (−1.77) | | |
| Control Variable | No | Yes | Yes | Yes | Yes | Yes |
| Company fixed effect | Yes | Yes | Yes | Yes | Yes | Yes |
| Year fixed effect | Yes | Yes | Yes | Yes | Yes | Yes |
| N | 669 | 533 | 533 | 557 | 550 | 550 |
| $R^2$ | 0.02 | 0.32 | 0.32 | 0.15 | 0.19 | 0.18 |

In conducting the multi-period DID regression analysis, two critical issues warrant attention. First, the control group is substantially smaller than the treatment group, which may result in estimates being disproportionately driven by information from the treatment group (MacKinnon & Webb, 2020 [40]). Second, even when employing large language models to screen digital transformation texts, the risk of greenwashing cannot be fully eliminated. These concerns underscore the necessity of incorporating exogenous shocks rather than relying solely on annual report texts.

To address this, we exploit the timing of digital transformation-related policy announcements as an exogenous shock. Specifically, following Tian and Zhang (2022) [41], we use the "Broadband China" policy as a proxy for the development of the digital economy. The network infrastructure upgrade of the "broadband China" pilot is a widely used exogenous policy shock reflecting digital transformation. Pilot cities and their corresponding implementation years are based on the website of the Ministry of Industry and Information Technology from 2014 to 2016.

Penalty texts contain company, penalty, and region details. We generate a city-year violation variable as the dependent variable and use "Broadband China" pilot cities and time for multi-period DID regression. Policy dummy equals 1 for pilot cities, 0 otherwise; time dummy equals 1 in post-treatment years and 0 otherwise. Figure A in the S1 File document shows the parallel trends test. The parallel trends assumption is supported, as the test passes with a two-period lag.

Our findings suggest that policy-induced digital transformation increases insurers' misconduct, reinforcing our earlier conclusion. Notably, the "Broadband China" policy provides an exogenous shock, affecting the digital infrastructure of local governments rather than the internal transparency of individual companies. If the development of the digital economy promoted by the policy leads to an increase in insurers' misconduct, and the effects have lagging, long-term, and growth effects (that is, misconduct does not rise immediately after the implementation of the policy, but increases year by year with the impact of insurer digitalization), it is more likely to be a change in corporate behavior itself, rather than simply because digitalization makes misconduct easier to detect. Due to space constraints, the multi-period DID results based on the policy shock are reported in the Table B in the S1 File document.

**4.2.2 Instrumental variable test.** The IV strategy mitigates potential endogeneity concerns and enhances estimation robustness. In this section, we construct a company-level insurer's technical adoption as an instrument variable, which is expressed by the number of patents. Companies with a higher level of technological innovation are more likely to engage in digital transformation (Gao et.al, 2025 [42]), while the number of patents is unlikely to directly affect insurers' misconduct. This satisfies the relevance and exogeneity conditions of a valid instrument. Using a two-stage least squares (2SLS) estimation, we isolate the exogenous variation in digital transformation driven by technology adoption, thereby mitigating potential endogeneity concerns.

We first obtain the published patent data of all applicants containing the word "insurance" from the Chinese patent database, including key information such as patent application number, applicant name, year of application, and patent type. We then screen patents filed by patent applicants for insurers and identify all insurer-related patents based on company names. To address discrepancies in company names, we cross-reference them with those in our panel dataset to ensure consistency. After that, patents filed by non-insurer entities (such as individual applications or non-insurance industry companies) are excluded, while invalid or withdrawn patent data are excluded to ensure data quality. Finally, according to the company-year dimension, we count the number of patents applied by each insurer in each year, and construct the patent number variable of "company-year" and match it with the panel data set. This variable is used to measure the accumulation of technology by insurers in a given year and as an instrumental variable to address the endogeneity of digital transformation.

After constructing the instrument variable, employ the two-stage least squares (2SLS) method for estimation. In the first stage, we use the instrumental variable as an explanatory variable to perform regression with the company-level digital transformation indicator. We find that the impact of technical adoption on digital transformation is significantly positively correlated, and the F-statistic of regression in the first stage is much larger than 10, which means that the problem of weak instrumental variables does not exist. In the second stage, digital transformation still shows a significant positive correlation with insurers' misconduct, indicating that the regression result is consistent with the baseline results after excluding endogenous interference. In addition, the result of the Sargan over-recognition test shows that the p-value does not significantly reject the null hypothesis and supports the externality of the instrumental variable.

**4.2.3. PSM method.** We also employ the propensity score matching (PSM) method to mitigate potential selection bias, recognizing that firms' digital transformation decisions are non-random and influenced by firm-specific characteristics. According to whether the value of the original explanatory variable is greater than 0, the sample insurers are divided into the treatment group and the control group, and the control variable in the baseline regression is used as the covariate to perform the nearest neighbor matching to achieve the regression estimation. The calculated mean treatment effect (ATT) of the treatment group is 14.9, indicating a significant and robust difference between the matched treatment group and the control group. In addition, the standard deviation is reduced to a low level. These results again show that our baseline study remains robust.

## 4.3. Heterogeneity analysis

Based on prior research, we aim to investigate whether the impact of digital transformation on insurers' misconduct differs across companies, periods, and types of digital transformation. We perform grouped regressions on sample data from four aspects.

**4.3.1. The impact of insurer types.** Adopting the methodology of de Bandt and Overton (2022) [43], we stratify the research sample by insurance category for heterogeneity analysis. The sample comprises 35 property insurers with 240 observations and 37 life insurers with 317 observations. Regression outcomes disclose that, in contrast to life insurers (Table 6, Column (2)), the misconduct of property insurers (Table 6, Column (1)) is more susceptible to the extent of their digital transformation. The underlying rationale might be the relatively short operating cycle of property insurance, along with its rapid product turnover, which curtails the time available for managing risks stemming from technological imperfections.

**4.3.2 The impact of capital types.** We partition the research sample into purely Chinese-funded and non-purely Chinese-funded (joint or foreign) insurers to conduct a heterogeneity analysis in light of corporate nature. The regression findings indicate that relative to purely Chinese-funded insurers (Table 6, Column (4)), the misconduct of joint or foreign insurers (Table 6, Column (3)) is more sensitive to the degree of their digital transformation. This could be attributed to the more intricate capital origins and management modalities of the latter, engendering additional compliance hurdles in the course of insurers' digital transformation.

**4.3.3 The impact of policy implementation.** Given the policy lag, we take the final year of the "Broadband China" pilot as the demarcation, splitting the research sample into two subsets: 2010–2016 and 2017–2021, for a heterogeneity analysis of policy impacts. Regression results indicate that prior to policy implementation, digital transformation's effect on insurers' misconduct is insignificant (Table 6, Column (5)). Post-implementation, however, it exerts a significant positive influence (Table 6, Column (6)). This suggests that the "Broadband China" initiative has substantially accelerated Chinese insurers' digital transformation. Yet, while digital applications offer new development prospects, they concurrently present considerable risk challenges, demanding heightened attention from all stakeholders.

**4.3.4. The impact of digital transformation types.** Based on the prior structural classification of the digital transformation dictionary, we extract the word frequency data of basic technology and technology development, and application in digital transformation, respectively. Taking their natural logarithms, we get indicators $digit1\_cluster1$ and $digit1\_cluster2$ for a heterogeneity analysis of digital transformation types. The regression results show that digital transformation in basic technology has a significantly negative impact on insurers' misconduct (Table 6, Column (7)), while that in technology development and application has a significantly positive one (Table 6, Column (8)). This indicates that introducing new technologies into the insurance industry does not bring compliance risk, but technology development and application beyond a company's management capacity can lead to misconduct and amplify compliance risk.

## 5. Further mechanisms analysis

### 5.1. Channel analysis

Prior research findings indicate that digital transformation in the insurance industry significantly increases the incidence of misconduct. This observation challenges the commonly held belief in the literature that digital transformation has a positive effect on operational efficiency, financial performance, and systemic risk management. To gain a deeper understanding of the risks of misconduct associated with insurers' digital transformation, this section employs channel analysis to systematically explore the underlying causes. This analysis reveals the inherent trade-off between the economic benefits of digital transformation and the associated risks of misconduct.

Following prior researches (Ding et al., 2024 [3]; Chen et al., 2023 [44]; Imran et al., 2025 [45]; Imran et al., 2025 [46]; Ding et al.,2025 [47]), we employ quantile regression to better capture the asymmetric effects of digital transformation on corporate misconduct. The results show that at lower quantiles (10%−30%), the coefficients are positive but insignificant, indicating limited impact for firms with low misconduct propensity. From the 40th quantile onward, the coefficients become statistically significant ($p < 0.05$) and exhibit a monotonic upward trend, increasing from 0.86 at the 40th quantile to 1.45 at the 70th, and reaching a maximum of 1.97 at the 90th quantile ($p < 0.1$). The results reveal a "risk amplification effect", whereby digital transformation intensifies misconduct risks among poorly governed firms. It indicates that digital

**Table 6. Heterogeneity Analysis.**

| Dependent Variable | | | | |
|---|---|---|---|---|
| Explanatory Variable | *sum_fine* | | | |
| | Insurer Type | | Capital Type | |
| | Property insurer | Life insurer | Joint or foreign | Purely Chinese-funded |
| | (1) | (2) | (3) | (4) |
| *digit*1 | 1.09* (1.94) | 0.22** (2.54) | 1.75*** (4.18) | 0.24* (1.90) |
| Control Variable | Yes | Yes | Yes | Yes |
| Company fixed effect | Yes | Yes | Yes | Yes |
| Year fixed effect | Yes | Yes | Yes | Yes |
| N | 240 | 317 | 356 | 201 |
| $R^2$ | 0.49 | 0.39 | 0.43 | 0.41 |

| Dependent Variable | | | | |
|---|---|---|---|---|
| Explanatory Variable | *sum_fine* | | | |
| | Policy Implementation | | Digital Transformation Type | |
| | 2010−2016 | 2017−2021 | Basic | Development & Application |
| | (5) | (6) | (7) | (8) |
| *digit*1 | -0.12 (−0.61) | 0.96* (1.85) | | |
| *digit*1_*cluster*1 | | | -5.27*** (−2.69) | |
| *digit*1_*cluster*2 | | | | 0.22*** (2.77) |
| Control Variable | Yes | Yes | Yes | Yes |
| Company fixed effect | Yes | Yes | Yes | Yes |
| Year fixed effect | Yes | Yes | Yes | Yes |
| N | 317 | 240 | 306 | 306 |
| $R^2$ | 0.29 | 0.16 | 0.20 | 0.24 |

transformation affects firms through specific mechanisms rather than uniformly increasing misconduct, thereby motivating a closer examination of the channels through which digital transformation shapes misconduct.

From an insurance practice perspective, insurers typically pursue digital transformation for two main reasons. First, to enhance competitiveness for a greater market share and a higher market position. Second, to optimize business models, reduce operating costs, and ease financial pressure. Alfiero et al. (2022) [48] and Srivastava et al. (2024) [27] noted that emerging technologies can expand marketing channels and improve service quality, boosting public trust and acceptance of insurers. However, a critical issue arises: The application barriers of these technologies may cause resource mismatch during digital transformation. The substantial financial and operational resources required for technological innovation may constrain insurers' capacity to strengthen compliance risk management. This resource allocation trade-off could exacerbate the "innovators' dilemma" (Christensen, 2015 [25]), wherein insurers prioritize digital transformation to expand premium revenues but inadvertently encourage risk-taking and opportunistic behavior. Simultaneously, the premium income increase from technological progress means more policies and greater administrative difficulty. If managers lack the expertise to allocate resources effectively (Bharadwaj et al., 2013 [49]), insurers' misconduct likelihood may rise. In other words, digital transformation may increase insurers' misconduct by influencing market expansion.

Moreover, digital technology application aids insurers in achieving smarter, online, and automated claims processing. Yet, existing technical loopholes, coupled with less stringent supervision and non-standardized regulatory rules, can significantly

increase the moral hazard of insurance holders and operational risk of staff. Consequently, insurers may incur more losses and financial pressure (Vidyavathi, 2013 [50]; Ben Dhiab, 2021 [51]), posing greater management challenges. How to relieve technology-related financial pressure is another route through which digital transformation may increase insurers' misconduct.

To validate the preceding analysis, we explore how digital transformation affects insurers' misconduct from revenue and cost angles. We introduce channel-related regression variables and construct the channel regression model, which is outlined below:

$$sum\_fine_{i,t} = \varphi_0 + \varphi_1 digit1_{i,t} + A \times controls_{i,t} + \mu_i + \gamma_t + \xi_{i,t} \tag{3}$$

$$Mediator_{i,t} = \theta_0 + \theta_1 digit1_{i,t} + A \times controls_{i,t} + \mu_i + \gamma_t + \xi_{i,t} \tag{4}$$

$$sum\_fine_{i,t} = \varphi'_0 + \varphi'_1 Mediator_{i,t} + \varphi'_2 digit1_{i,t} + A \times controls_{i,t} + \mu_i + \gamma_t + \xi_{i,t} \tag{5}$$

Where $Mediator_{i,t}$ is the channel regression variable, which includes loss ratio ($Lr$) and premium income ($PI$). Citing Born et al. (2023) [52] and Ritho et al. (2023) [53], we take premium income and loss ratio (Claims Incurred/Premiums Earned) as indicator variables to explore digital transformation's impact on insurers' misconduct via market expansion and financial pressure channels. Data are from the "China Insurance Yearbook". Other notations remain as above.

Notably, given the academic disputes over channel analysis, we further test the authenticity of channel variables and the effectiveness of channel effects. Firstly, we test the significance of digital transformation on channel variables. Secondly, without considering channel effects, we test the significance of digital transformation on insurers' misconduct. Thirdly, we test the significance of channel variables on insurers' misconduct. Fourthly, considering channel effects, we check if the impact of digital transformation on insurers' misconduct is reduced compared to when ignoring them. The results are summarized in Table 7 and Table 8. It shows that variables of both income and cost channels passed the Sobel mediation test, indicating the high credibility of our analysis.

**5.1.1. Market expansion channel.** Observing Columns (1)-(2) in Table 7 reveals that digital transformation increases insurers' misconduct via boosting premium income. Digital transformation enables insurers to explore more marketing channels and offer smarter, targeted, and inclusive services to policyholders (Alfiero et al., 2022 [48]; Srivastava et al., 2024 [27]), thus hiking insurance income. However, as per Christensen's (2015) [25] "Innovator's Dilemma" theory, with unchanged human and physical capital, innovative services consume vast resources and heighten management challenges. Consequently, resource constraints expose insurers to more misconduct risk.

Results in Table 8 indicate that the indirect effect of digital transformation on insurers' misconduct through the market expansion channel accounts for 18.4% of the total effect.

**5.1.2. Financial pressure channel.** Observing Columns (3)-(4) in Table 7 shows that digital transformation boosts insurers' misconduct by increasing loss ratios. Essentially, digital transformation simplifies claim submissions for customers, heightening risks of false or excessive claims and upping the loss ratio. A higher loss ratio then burdens insurers with greater financial stress, impairing liquidity and profitability (Vidyavathi, 2013 [50]; Ben Dhiab, 2021 [51]). Strapped for cash, insurers might cut costs via non-compliant means like flawed claim reviews or lowered settlement standards, escalating misconduct risk.

Table 8 results indicate that the indirect effect of digital transformation through the financial pressure channel accounts for 28.2% of the total effect on insurers' misconduct.

## 5.2. Complementary tests of plausible channels

In the previous section, we examine why insurers undergoing digital transformation may engage in misconduct more frequently despite the risks of detection and subsequent penalties. The above analysis shows that market expansion and

**Table 7. Channel Regression and Test Results.**

| Dependent Variable | | | | |
|---|---|---|---|---|
| Explanatory Variable | Market Expansion Channel | | Financial Pressure Channel | |
| | *PI* | *sum_fine* | *Lr* | *sum_fine* |
| | (1) | (2) | (3) | (5) |
| *digit*1 | 0.19*** (4.30) | 0.88*** (2.95) | 6.03*** (4.45) | 0.79*** (2.67) |
| PI | | 0.89*** (3.05) | | |
| Lr | | | | 0.04*** (4.60) |
| Control Variable | Yes | Yes | Yes | Yes |
| Company fixed effect | Yes | Yes | Yes | Yes |
| Year fixed effect | Yes | Yes | Yes | Yes |
| Sobel | —— | 0.16** | —— | 0.25*** |
| N | 533 | 533 | 532 | 532 |
| $R^2$ | 0.87 | 0.34 | 0.17 | 0.36 |

**Table 8. Direct Effects, Indirect Effects and Total Effects of Channel Variables and Their Relative Relations.**

| Mediating variables | | Observed Coef. | Std_err | z | P>z |
|---|---|---|---|---|---|
| PI | Indirect effect | 0.163 | 0.064 | 2.535 | 0.011 |
| | Direct effect | 0.725 | 0.294 | 2.468 | 0.014 |
| | Total effect | 0.888 | 0.293 | 3.033 | 0.002 |
| Lr | Indirect effect | 0.251 | 0.08 | 3.144 | 0.002 |
| | Direct effect | 0.638 | 0.292 | 2.185 | 0.029 |
| | Total effect | 0.889 | 0.293 | 3.03 | 0.002 |
| Proportion | | | | **PI** | **Lr** |
| Proportion of the total effect that is mediated | | | | 0.184 | 0.282 |
| Ratio of indirect to direct effect | | | | 0.225 | 0.393 |
| Ratio of total to direct effect | | | | 1.225 | 1.393 |

financial pressure are two plausible channels digital transformation affects insurers' misconduct. Here, we look into the moderating mechanisms of this impact and offer more proof for the prior channel analysis. Specifically, we pick factors that affect insurers' market expansion and financial pressure from internal operations and external environment angles. Then, we analyze how these factors influence the link between digital transformation and insurers' misconduct.

From the internal operation perspective, on the one hand, an insurer's current market share is a key indicator of its market competitiveness and a major factor influencing its market expansion. Insurers with higher market share generally have greater market expansion potential and are better positioned to leverage digital technologies to strengthen their market position. Nevertheless, allocating too many resources to market expansion may lead to neglect of compliance management, thereby worsening misconduct. On the other hand, commission fees and operating expenses are significant components of insurers' cost outlays and important factors influencing their financial pressures. When insurers reduce investments in handling fees, commission fees, and operating expenses, their employees have lower salary incentives to adopt compliant marketing and operations, and may pursue interests more aggressively. Consequently, in the digital transformation process of insurers, some insurer staff might resort to radical work strategies, resulting in improper behaviors such as misleading sales, bundling, and illegal profit-making.

From the external environment perspective, insurance density and penetration are key indicators of a region's insurance industry maturity. The market supervision system is more robust in areas with mature insurance industries, and the external pressure to constrain insurers' misconduct is greater. Even if insurers have strong market expansion motives during digital transformation, they are less likely to resort to illegal methods. Moreover, in insurance markets with higher overall premium income growth rates, insurers have ample market resources to expand their customer base through digital technologies. This can help reduce malpractices driven by cutthroat competition (Kumaraswamy et al., 2018 [54]). On the other hand, higher economic uncertainty introduces more instability to insurers' operations, leading to increased financial pressures and more frequent misconduct. Additionally, a more strictly regulated insurance market raises the detection rate of insurers' misconduct, yet it also indirectly pushes insurers to strengthen compliance management. How regulatory intensity modulates the impact of digital transformation on insurers' misconduct depends on the combined effect of these two opposing directions.

Based on the above analysis, relevant indicators are selected to construct the mechanism test model as shown below.

$$sum\_fine_{i,t} = \alpha_0 + \alpha_1 digit1_{i,t} \times MVs_{i,t} + A \times controls_{i,t} + \mu_i + \gamma_t + \xi_{i,t} \tag{6}$$

Where $MVs_{i,t}$ comprises market share ($MS$), handling fees & commission expenses ($Comm$), operating expenses ($Oper$), insurance density ($Pene$), insurance penetration ($Den$), national premium income growth rate ($InGr$), policy uncertainty index ($EPU$), and insurance supervision intensity ($Regu$), etc. Variable symbols, construction methods, data sources, and references of the mechanism test model are summarized in Table 9. Other notations remain as above.

### 5.2.1. Regulating effect of internal operation.
Referring to Alexander and Neill (2015) [55], Wang et al. (2020) [56] and Nanda and Gopalaswamy (2024) [57], we use market share (Table 10, Column (1)), handling fees & commission expenses (Table 10, Column (2)) and operating expenses (Table 10, Column (3)) to explore how internal operations moderate digital transformation's impact on insurers' misconduct.

We find that the interaction of digital transformation and market share positively affects misconduct, while those of digital transformation with handling fees and commission expenses and operating expenses have negative impacts. In other words, insurers with higher market share and lower handling fees and commissions, and operating expenses are more likely to violate rules during digital transformation. These empirical findings provide actionable insights applicable to contemporary insurance market dynamics.

Digital transformation significantly enhances operational efficiency and market penetration through streamlined business processes and expanded distribution channels, thereby strengthening insurers' competitive positioning and market share. However, this market expansion creates a paradoxical effect: Increasing policy volumes amplify operational complexity, heightening the risk of governance failures. Large-market-share insurers often prioritize revenue growth over organizational adaptation, reallocating resources from digital governance to sales initiatives. This misalignment impairs their capacity to manage technological changes during system upgrades, leading to higher non-compliance rates as employees struggle with evolving protocols.

Concurrently, high R&D expenditures for digital infrastructure compress profit margins, incentivizing cost-cutting measures such as reducing compliance budgets or misclassifying expenses. However, immature technological applications compound this compliance management pressure by introducing systemic vulnerabilities. For instance, incomplete audit trails and inconsistent data standards in legacy systems create opportunities for financial manipulation. Additionally, the monetizable value of customer data stored in digital platforms increases the likelihood of insider threats, with employees potentially exploiting access privileges for illicit gains. These converging factors create an environment where regulatory compliance becomes increasingly challenging, undermining long-term sustainability.

The key implication is that insurers must balance market expansion strategies with commensurate investments in governance frameworks. Prioritizing internal controls and employee training during digital adoption can mitigate operational risk, aligning technological innovation with ethical business practices.

**Table 9. Mechanism Test Variable.**

| Symbols | Construction Method | Data Source | Reference |
|---|---|---|---|
| *MS* | *PremiumIncomeoftheCompany÷ TotalPremiumIncomeoftheIndustry* | China Insurance Yearbook | Alexander & Neill (2015) [55] |
| *Comm* | *CommissionExpenses + HandlingFees* | | Wang et al. (2020) [56] |
| *Oper* | *OperatingExpenses* | | Nanda & Gopalaswamy (2024) [57] |
| *Pene* | *TotalPremiumWritten÷ GrossDomesticProduct* | | Born & Bujakowski (2021) [58] |
| *Den* | *TotalPremiumWritten ÷ TotalPopulation* | | |
| *InGr* | *∆Premium Income ÷ Premium Income* | | |
| *EPU* | *Uncertainty Mentions in News Articles÷ Total Number of News Articles* | China Stock Market and Accounting Research Database | Peng et al. (2021) [59] |
| *Regu* | *FinancialRegulationExpenditure÷ FinancialSectorValueAdded* | State Statistics Bureau | Ge & Li (2020) [60] |

**Table 10. Mechanism Analysis.**

| **Dependent Variable** | | | | |
|---|---|---|---|---|
| **Explanatory Variable** | ***sum_fine*** | | | |
| | **(1)** | **(2)** | **(3)** | **(4)** |
| *digit*1 × *MS* | 1.16*** (6.20) | | | |
| *digit*1 × *Comm* | | -0.01** (−2.56) | | |
| *digit*1 × *Oper* | | | -0.01*** (−5.46) | |
| *digit*1 × *Dens* | | | | -3.89** (−2.36) |
| Control Variable | Yes | Yes | Yes | Yes |
| Company fixed effect | Yes | Yes | Yes | Yes |
| Year fixed effect | Yes | Yes | Yes | Yes |
| N | 533 | 546 | 554 | 557 |
| $R^2$ | 0.25 | 0.16 | 0.41 | 0.59 |

| **Dependent Variable** | | | | |
|---|---|---|---|---|
| **Explanatory Variable** | ***sum_fine*** | | | |
| | **(5)** | **(6)** | **(7)** | **(8)** |
| *digit*1 × *Pene* | -1.21* (−1.75) | | | |
| *digit*1 × *InGr* | | -5.77** (−2.17) | | |
| *digit*1 × *EPU* | | | 0.87*** (4.07) | |
| *digit*1 × *Regu* | | | | 0.12 (0.10) |
| Control Variable | Yes | Yes | Yes | Yes |
| Company fixed effect | Yes | Yes | Yes | Yes |
| Year fixed effect | Yes | Yes | Yes | Yes |
| N | 557 | 557 | 557 | 557 |
| $R^2$ | 0.58 | 0.59 | 0.59 | 0.13 |

**5.2.2 Regulatory effect of the external environment.** Referring to Born & Bujakowski (2021) [58], Peng et al. (2021) [59], and Ge and Li (2020) [60], we choose insurance density (Table 10, Column (4)), insurance penetration (Table 10, Column (5)), premium income growth (Table 10, Column (6)), economic uncertainty index (Table 10, Column (7)), and regulatory intensity (Table 10, Column (8)) to examine how the external environment moderates digital transformation's impact on insurers' misconduct. As these external environmental variables exist at the provincial level (province-year), we link them to firm-level data by matching each firm's headquarters location before conducting the moderation analysis.

Our research reveals that the interaction terms of digital transformation with insurance density, insurance penetration, and premium income growth rate all significantly and negatively affect misconduct. However, the interaction term of digital transformation and economic uncertainty significantly and positively impacts misconduct. Meanwhile, the interaction term of digital transformation and regulatory intensity shows no significant effect. In short, higher insurance density, penetration, and premium income growth can mitigate the negative effects of digital transformation on insurers' misconduct. In contrast, increased economic uncertainty exacerbates such misconduct, and changes in regulatory intensity don't significantly moderate digital transformation's influence on misconduct. In line with the actual insurance operation, the relevant phenomena can be interpreted as follows.

Firstly, growth in insurance density and penetration rates reflects evolving societal trust in financial intermediaries, while expanding premium income underscores the sector's capacity to scale operations. Collectively, these trends suggest strengthened market positioning and operational capabilities, enabling insurers to better mitigate risks associated with digital transformation and thereby reduce unethical practices.

Secondly, during periods of macroeconomic volatility, firms face intensified operational stress, often prompting cost-cutting measures. Paradoxically, digital innovation necessitates sustained investment in R&D and infrastructure, creating a resource allocation dilemma. This tension between short-term survival imperatives and long-term innovation requirements may incentivize misconduct behavior as firms balance competing demands.

Finally, China's insurance industry operates under a historically strict regulatory framework, where incremental changes in enforcement intensity exert limited influence on compliance systems. Consequently, the relationship between digital transformation and misconduct remains invariant to regulatory shifts, indicating that observed behavioral changes stem from endogenous firm-level adaptations rather than exogenous regulatory detection effects.

# 6. Conclusion and discussion

This paper provides novel evidence on the relationship between digital transformation and insurers' misconduct as reflected in regulatory penalties. Digital transformation amplifies misconduct in ways that systematically influence insurers' regulatory compliance, rather than representing a mere technological upgrade. To measure digital transformation in the insurance industry, we construct a novel dictionary combining industry-specific and general digital-related terms, and apply text analysis to annual information disclosure reports of Chinese insurers.

The findings are as follows: First, digital transformation increases insurers' misconduct, and this result is confirmed by robustness and endogeneity tests. Second, property insurers and those with joint or foreign ownership are more affected by digital transformation in terms of misconduct. The "Broadband China" policy strongly drives digital transformation but also exacerbates misconduct. While adopting new technologies is not inherently risky, the misalignment between technological applications and managerial capabilities contributes to misconduct. Third, digital transformation drives insurers' misconduct through market expansion and financial pressure channels. Finally, insurers with higher market share, lower commissions, and reduced operational expenses are more prone to misconduct during the digital shift. Conversely, higher insurance density, greater penetration, and stronger premium growth help mitigate the negative effect of digital transformation on compliance. Economic uncertainty worsens misconduct, while changes in regulatory intensity do not exert a significant moderating effect.

China's experience demonstrates that while digitalization often improves insurers' profitability, it may also create incentives for misconduct that are frequently overlooked. This tension is especially salient for Asian and Latin American countries grappling with analogous challenges of limited insurance coverage and accelerated fintech proliferation. Therefore, understanding the misconduct risk during insurers' digital transformation is crucial, as it directly affects industry integrity and the sustainability of digital innovation. Existing research largely emphasizes the efficiency and performance gains from digital transformation but pays limited attention to its unintended risks.

Our findings offer key implications for insurers and regulators. For insurers, digital transformation should be accompanied by strengthened internal risk management. Standardized practices in underwriting and claims oversight are essential to curb excessive risk-taking, particularly among property insurers and joint or foreign firms. For regulators, the results underscore the importance of adopting a targeted, risk-based supervisory approach, rather than merely intensifying regulation during digitalization. In addition, promoting the overall development of insurance markets can help mitigate the misconduct risk associated with digital transformation.

Our research highlights the unforeseen risks associated with insurers' digital transformation, thereby enriching the existing literature. Nevertheless, several limitations merit further exploration. First, our results suggest that insurer type and capital structure shape heterogeneous effects of digital transformation on misconduct, likely due to differences in business processes. However, the complexity and limited observability of these processes prevent us from fully incorporating this dimension into the empirical framework. Second, while our findings indicate that digital transformation expands market scale and increases financial stress through rising premiums and loss ratios, questions regarding how insurers can effectively balance market expansion, cost control, and compliance management remain insufficiently addressed due to space constraints. Future research could extend this study in several directions. One avenue would be to investigate the broader implications of digital transformation for financial regulation, corporate governance, and consumer confidence. Particular attention should be paid to how regulators can leverage digital technologies to enhance supervisory effectiveness and how insurers can strengthen compliance management efficiency during the digital transition. Such efforts will contribute to a more comprehensive understanding of the multifaceted impacts of digital transformation on the insurance industry. In addition, the quantile regression results indicate that for insurers exposed to higher misconduct risks, digital transformation further amplifies propensity for misconduct, suggesting that it may create new opportunities for regulatory arbitrage. Future research could explore this issue in light of this asymmetric effect.

## Supporting information

**S1 File. Supporting Information is "S1", which includes the Table A, Table B, Figure A,and Data.**
(XLSX)

## Author contributions

**Data curation:** Xiaoqing Guo.

**Formal analysis:** Xiaoqing Guo.

**Investigation:** Jiandi Zhang.

**Methodology:** Jiandi Zhang, Xiaoqing Guo.

**Project administration:** Jiandi Zhang, Zhengfa Yang.

**Software:** Xiaoqing Guo.

**Supervision:** Rui Xu, Zhengfa Yang.

**Validation:** Jiandi Zhang, Xiaoqing Guo.

**Writing – original draft:** Jiandi Zhang.

**Writing – review & editing:** Jiandi Zhang, Xiaoqing Guo, Rui Xu.

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
