## [Decision Letter · Decision Letter 0]

15 Jul 2025

Dear Dr. Guo,

Thank you for submitting your manuscript to PLOS ONE. After careful consideration, we feel that it has merit but does not fully meet PLOS ONE’s publication criteria as it currently stands. Therefore, we invite you to submit a revised version of the manuscript that addresses the points raised during the review process.

We look forward to receiving your revised manuscript.

Kind regards,

Haijun Yang, Ph.D

Academic Editor

PLOS ONE

Journal Requirements:

2. Please note that your Data Availability Statement is currently missing [the repository name and/or the DOI/accession number of each dataset OR a direct link to access each database]. If your manuscript is accepted for publication, you will be asked to provide these details on a very short timeline. We therefore suggest that you provide this information now, though we will not hold up the peer review process if you are unable.

Additional Editor Comments (if provided):

Dr. Guo,

The reviewers and editors handling your paper have recommended that your paper undergo major revision. If you care to revise it, we will reconsider it for publication.

Best,

Yang

Reviewers' comments:

Reviewer's Responses to Questions

**Comments to the Author**

1. Is the manuscript technically sound, and do the data support the conclusions?

Reviewer #1: Yes

Reviewer #2: Partly

2. Has the statistical analysis been performed appropriately and rigorously?

Reviewer #1: Yes

Reviewer #2: Yes

3. Have the authors made all data underlying the findings in their manuscript fully available?

Reviewer #1: Yes

Reviewer #2: Yes

4. Is the manuscript presented in an intelligible fashion and written in standard English?

Reviewer #1: Yes

Reviewer #2: No

Reviewer #1: The paper discusses the violation risks in the digital transformation of the insurance industry, filling the gap in the existing literature that emphasizes efficiency over risk. The article reveals the positive correlation between digitalization and violations through Chinese market data and puts forward some policy suggestions. Although the topic selection is quite meaningful and the method is rigorous, there are still some areas that need improvement.

The introduction section follows the common presentation style of introducing the research background, identifying literature gaps, posing research questions, and elaborating on research value. However, elaborating on research value seems too long. It is best to concisely point out your research value in the introduction section, so that readers can understand it at a glance and quickly capture your innovation points.

2. Your literature review section seems to be combined with the introduction, followed by the sources of data and the selection of variables. However, a complete causal analysis should be conducted by applying economic theories before officially starting the quantitative analysis to provide theoretical support for your quantitative analysis. But I did not find this part in your article.

3. There seem to be some grammatical inappropriateness. For example, the article mentions insurer misconduct many times, and it seems more appropriate to change it to insurers' misconduct, and try to maintain consistent expressions for technical terms.

4. Although your robustness test is very rich, it seems to have not touched upon the essence. The digital transformation of insurance companies has extensively used the frequency of annual report words, but the annual report text may have the phenomenon of "greenwashing", leading to measurement bias. It is hoped that a robustness test targeting this issue can be added.

4. In terms of literature citation, it is advisable to cite literature with the latest year as much as possible, such as Manning, C. D. (1999). Such as the literature is too old, it is recommended that the reference https://doi.org/10.1016/j.envres.2023.118074 and https://doi.org/10.1080/09640568.2024.2311821 these two articles, The journals they are in are all relatively authoritative and updated annually.

Reviewer #2: Thank you for your efforts on this important topic. While the paper presents a rich dataset and thoughtful analysis, there are major concerns that need to be addressed. The theoretical framework is underdeveloped and needs a clearer structure and depth. The way you measure key variables, particularly digital transformation and misconduct, requires stronger justification and validation. Several interpretations seem one-sided, leaning too heavily on confirming the hypothesis without engaging with alternative explanations. Finally, the manuscript would benefit from thorough language editing to improve clarity and professionalism.

**Do you want your identity to be public for this peer review?** For information about this choice, including consent withdrawal, please see our Privacy Policy

Reviewer #1: No

Reviewer #2: No

---

## [Author Response · Author response to Decision Letter 1]

27 Aug 2025

Dear Editors and Reviewers:

We sincerely thank the editor for providing us with the valuable opportunity to revise the manuscript, and we also deeply appreciate the two reviewers for devoting their time and effort to offering detailed comments. These comments have greatly clarified the direction for revision and helped us improve the quality of the manuscript. We have carefully considered them and revised the manuscript comprehensively. In response to the issues raised by the reviewers, we hereby provide point-by-point replies. Specifically, these are as follows:

1. The revision explanation for the Reviewer's Responses to Questions

Question 1. Is the manuscript technically sound, and do the data support the conclusions? [The manuscript must describe a technically sound piece of scientific research with data that supports the conclusions. Experiments must have been conducted rigorously, with appropriate controls, replication, and sample sizes. The conclusions must be drawn appropriately based on the data presented.] Reviewer #1: Yes, Reviewer #2: Partly.

Response: Thank you for the valuable suggestion. Under your guidance, we systematically reviewed the experimental design. We supplemented the analysis with new robustness test indicators based on higher data collection standards and conducted regression analyses. Additionally, we constructed a multi-period difference-in-differences model based on the "Broadband China" policy to corroborate the research conclusions. These revisions help to avoid inappropriate inferences, thereby strengthening the technical rigor of the study and enhancing the reliability of the conclusions.

Revision Content:

(1) The paragraphs 3-6 of the “4.1 Benchmark regression analysis and Robust test”:

“……We utilize text analysis to extract data on the number of penalties from the NFRA ().

To address potential greenwashing in annual reports, whereby insurers may exaggerate or even fabricate their digital transformation efforts, we train and fine-tune a large language model (LLM) to further filter digital terms. The model is designed to accurately distinguish between action-level and non-action-level statements. For example, a sentence containing a digital transformation term is retained only if it explicitly describes a concrete action undertaken by the firm to advance digital transformation. This approach preserves the applicability of text-based measures while mitigating bias arising from greenwashing, thereby providing a more accurate reflection of firms’ actual progress in digital transformation. Our objective is to minimize the impact of greenwashing, albeit at the risk of underestimating the extent of digital transformation.

The construction proceeds as follows. First, we label all sentences containing digital transformation terms and identify the corresponding firm names and report years. Second, we use a bootstrap sampling method to select 1,000 sentence pairs for manual annotation, and employ a custom-designed prompt to train the LLM. Third, after training, we classify each sentence in the sample according to its semantic level and retain only keywords identified as “action-level” to construct the final digital transformation measure (). The LLM is fine-tuned based on Qwen2.5, achieving over 95% classification accuracy on the validation set.

In robustness tests, we use the LLM-based digital transformation measure () and the number of regulatory penalties () as alternative independent and dependent variables, respectively……”

(2) Paragraphs 5-8 of the “4.2.1 Multi-Period DID Model”:

“……Second, even when employing large language models to screen digital transformation texts, the risk of greenwashing cannot be fully eliminated. These concerns underscore the necessity of incorporating exogenous shocks rather than relying solely on annual report texts.

To address this, we exploit the timing of digital transformation-related policy announcements as an exogenous shock. Specifically, following Tian and Zhang (2022), we use the “Broadband China” policy as a proxy for the development of the digital economy. The network infrastructure upgrade of the "broadband China" pilot is a widely used exogenous policy shock reflecting digital transformation. Pilot cities and their corresponding implementation years are based on the website of the Ministry of Industry and Information Technology from 2014 to 2016.

Penalty texts contain company, penalty, and region details. We generate a city-year violation variable as the dependent variable and use “Broadband China” pilot cities and time for multi-period DID regression. Policy dummy equals 1 for pilot cities, 0 otherwise; time dummy equals 1 in post-treatment years and 0 otherwise. Figure A in the S1 document shows the parallel trends test. The parallel trends assumption is supported, as the test passes with a two-period lag.

Our findings suggest that policy-induced digital transformation increases insurers’ misconduct, reinforcing our earlier conclusion. Notably, the "Broadband China" policy provides an exogenous shock, affecting the digital infrastructure of local governments rather than the internal transparency of individual companies. If the development of the digital economy promoted by the policy leads to an increase in insurers’ misconduct, and the effects have lagging, long-term, and growth effects (that is, misconduct does not rise immediately after the implementation of the policy, but increases year by year with the impact of insurer digitalization), it is more likely to be a change in corporate behavior itself, rather than simply because digitalization makes misconduct easier to detect. Due to space constraints, the multi-period DID results based on the policy shock are reported in the Table B in the S1 document.”

(3) Column (6) in Table 4.

Question 4. Is the manuscript presented in an intelligible fashion and written in standard English? [PLOS ONE does not copyedit accepted manuscripts, so the language in submitted articles must be clear, correct, and unambiguous. Any typographical or grammatical errors should be corrected at revision, so please note any specific errors here.] Reviewer #1: Yes, Reviewer #2: No.

Response and Revision Content: Thank you for the valuable suggestion. Under your guidance, we carefully refined the language throughout the manuscript, corrected formatting and grammatical errors, and improved the clarity and coherence of expression. We strived to ensure precise and accurate language that conforms to standard academic English norms, thereby enhancing the overall readability of the manuscript.

Other questions: The responses from the reviewers are all "Yes".

2. Explanation of Revisions for the Review Comments to the Author

2.1 Explanation of Revisions for Reviewer 1’s Comments

We greatly appreciate Reviewer 1's recognition of the research content, methodology, and significance of the manuscript, as well as the valuable suggestions for revision. In response to each issue raised, we have made corresponding revisions and provide the following point-by-point replies.

Comment 1. The introduction section follows the common presentation style of introducing the research background, identifying literature gaps, posing research questions, and elaborating on research value. However, elaborating on research value seems too long. It is best to concisely point out your research value in the introduction section, so that readers can understand it at a glance and quickly capture your innovation points.

Response: Thank you for the valuable suggestion. Under your guidance, we have comprehensively reviewed the content of the “1. Introduction” section, simplifying it as much as possible while ensuring complete expression without altering the original meaning. We paid special attention to condensing the discussion of research value and highlighting the key points of the innovations. In addition, we also streamlined the general descriptions of the research content, methods, and conclusions.

(1)Research value: We reduced the originally lengthy five paragraphs to a more concise three paragraphs.

(2)Innovations: We summarized the key points in the first sentence of each innovation to enable readers to quickly grasp the main contributions.

(3) Research content and methods: We removed expressions that were highly repetitive with later sections and adopted general descriptions for a more concise presentation.

Revision Content:

(1) The paragraphs 1-3 of the “1. Introduction”:

“In the global insurance industry, the rapid embrace of digital technologies has significantly reshaped key operational areas, including product marketing, intelligent underwriting, automated claims processing, and fraud detection (Gatteschi et al., 2018; de Andrés & Gené, 2024). While the benefits of technological progress and digitalization are well-documented (Ding et al., 2024; Zhang et al., 2024), their early adoption phase may also open new channels for misconduct and regulatory violations.

Enterprise misconduct is a distinct operational risk, defined as unethical business operations that violate laws or harm stakeholders (Xia et al., 2023). The Chinese insurance market provides a representative case for examination. Regulatory data from China indicate a rise in insurers’ misconduct over the past decade, coinciding with a period of rapid digital transformation in the insurance industry[Since the State Council launched the “Broadband China” plan in 2013, digital transformation in China has accelerated. Concurrently, data from the China Financial Regulatory Administration show that instances of insurers’ misconduct have increased alongside advancements in digital technology. Specifically, there has been a rapid increase in penalties for misconduct since 2017. In 2024, Chinese insurers received 2,762 misconduct penalties, with total fines amounting to 280 million yuan, representing an increase of 137.3% compared with 2017.]. Therefore, the following questions are raised: Are there risks of misconduct in the digital transformation of insurers? Why does the digital transformation of insurers present new challenges in managing misconduct while advancing the industry?

However, existing literature has predominantly emphasized the positive impacts of digitization on market value (Fritzsch et al., 2021), sales innovation (Eckert et al., 2021), product customization (Radwan, 2019), pricing precision (Radwan, 2019), customer service optimization (e Sá et al., 2024), and fraud reduction (Radwan, 2019), with limited attention to potential risks, especially misconduct ones. Consequently, understanding whether digital transformation in the insurance sector is associated with an increase in misconduct contributes to corporate governance and regulatory oversight, especially in light of the growing prevalence of digital technologies among insurers.”

(2) The paragraphs 8-10 of the “1. Introduction”:

“……First, we construct measurement indicators for digital transformation and misconduct through test analysis for insurers……

……Second, we highlight the often-overlooked risks of digital transformation……

……Third, we investigate the mechanism underlying the impact of digital transformation……”

(3) The paragraphs 4-6 of the “1. Introduction”:

“Using data from Chinese insurers between 2010 and 2021, this empirical study is the first to provide evidence on how digital transformation influences insurers’ misconduct and the underlying mechanisms in an evolving insurance market. We compile annual disclosure reports data from 72 insurers in China (the majority of which are non-listed) and construct a dictionary for digital transformation through text analysis. Next, following Raghunandan (2024), we extract penalty amounts and fines imposed on insurers from the violation announcements issued by the regulator[ Using government penalty data to construct the indicator of insurers’ misconduct presents a potential controversy: Does the rise in government-imposed penalties reflect an actual increase in corporate misconduct, or is it merely a result of heightened regulatory scrutiny and an increased likelihood of detection during digitalization? To validate the appropriateness of our variable selection, we employ a range of methods, including the incorporation of interaction terms, the substitution of dependent variables, and quasi-natural experiments. The empirical analysis demonstrates that digital transformation indeed exacerbates insurers’ propensity to engage in misconduct.]. Based on the construction of the core variables, we introduce additional control variables and employ firm and year fixed effects to conduct an empirical analysis of the impact of digital transformation on insurers’ misconduct. The results suggest that digital transformation increases insurers’ misconduct, and robustness as well as endogeneity tests confirm the validity of these findings.

We then conduct a heterogeneity analysis by insurer type, capital structure, the implementation of the “Broadband China” policy, and the digital transformation category. The results show that property insurers and those with joint or foreign ownership are more affected by digital transformation regarding misconduct. The “Broadband China” policy accelerates insurers’ digital transformation but also exacerbates misconduct. While basic technological advancements in digital transformation help reduce misconduct, more advanced technology development and application have the opposite effect.

Furthermore, we construct a channel regression model, employing premium income and the cost ratio as proxies for market expansion and financial pressure variables, respectively. The result shows that the digital transformation of insurers increases misconduct by increasing premium income and cost ratio. In addition, we identify moderate variables to explore complementary tests as as evidence of plausible channels. Internally, firms with greater market power face stronger incentives for misconduct, whereas lower handling fees and operating expenses constrain opportunistic behavior. Externally, mature insurance markets curb misconduct, while economic uncertainty exacerbates it. However, changes in regulatory intensity do not significantly mitigate misconduct, suggesting that regulatory improvements alone cannot fully explain the rise in penalties.”

Comment 2. Your literature review section seems to be combined with the introduction, followed by the sources of data and the selection of variables. However, a complete causal analysis should be conducted by applying economic theories before officially starting the quantitative analysis to provide theoretical support for your quantitative analysis. But I did not find this part in your article.

Response: Thank you for the valuable suggestion. Under your guidance, we have reorganized the literature review, separating it from “1. Introduction” into an independent chapter titled “2. Literature Review and Hypotheses Development.” In this chapter, we supplemented several relevant economic theories, including resource allocation theory, deterrence theory, and information asymmetry theory, to provide a comprehensive causal analysis and theoretical support for the subsequent quantitative analysis.

Revision Content:

“2. Literature Review and Hypotheses Development

Enterprise misconduct has long been a significant barrier to the sustainability of the insurance industry and the sound functioning of insurers (Schiro, 2006). Scholars have identified a strong connection between insurers' internal management and misconduct (Rizwan, 2019; Ben, 2024; Gunaseelan et al., 2024). Some argue that changes in the external regulatory environment impact insurers’ misconduct (Chen & Hieber, 2016; Srbinoski et al., 2022; Koziol & Kuhn, 2023). Others have explored the consequences of misconduct for insurers and the broader industry (Makau et al., 2021; Talesh & Filho, 2023). Despite extensive research on company management, industry supervision, and economic outcomes, there is a notable gap in the literature regarding the impact of digital transformation on insurers’ misconduct, especially as technologies such as big data, cloud computing, Internet of Things (IoT), blockchain, and artificial intelligence (AI) continue to evolve and integrate into the sector.

---

## [Decision Letter · Decision Letter 1]

10 Sep 2025

Dear Dr. Guo

Thank you for submitting your manuscript to PLOS ONE. After careful consideration, we feel that it has merit but does not fully meet PLOS ONE’s publication criteria as it currently stands. Therefore, we invite you to submit a revised version of the manuscript that addresses the points raised during the review process.

We look forward to receiving your revised manuscript.

Kind regards,

Haijun Yang, Ph.D

Academic Editor

PLOS ONE

Journal Requirements:

Additional Editor Comments:

Dear Authors,

Please respond to the reviewer1's comments.

Best

Yang

Reviewers' comments:

Reviewer's Responses to Questions

**Comments to the Author**

Reviewer #1: (No Response)

Reviewer #2: (No Response)

2. Is the manuscript technically sound, and do the data support the conclusions?

Reviewer #1: (No Response)

Reviewer #2: Yes

3. Has the statistical analysis been performed appropriately and rigorously?

Reviewer #1: (No Response)

Reviewer #2: Yes

4. Have the authors made all data underlying the findings in their manuscript fully available?

Reviewer #1: (No Response)

Reviewer #2: Yes

5. Is the manuscript presented in an intelligible fashion and written in standard English?

Reviewer #1: (No Response)

Reviewer #2: Yes

Reviewer #1: This paper examines the impact of misconduct in insurers’ digital transformation. It is an insightful and valuable study. Before final publication, I hope the authors will consider the following points.

The mechanism section is not yet sufficiently developed. I recommend strengthening it by drawing on the following five articles:

https://doi.org/10.1016/j.envres.2023.118074

https://doi.org/10.1007/s11356-023-30307-z

https://doi.org/10.1016/j.esr.2024.101590

https://doi.org/10.1016/j.esr.2025.101837

doi: 10.2166/wp.2025.355

The introduction could more clearly articulate the research motivation—for example, why focus on China rather than the Middle East or South America, and why are samples from African regions important?

The implications section could delve deeper into the study’s international relevance—for instance, what lessons might China’s experience offer for resource policies in developing countries across Asia and South America?

The introduction should provide a more in-depth exposition of the research questions.

The manuscript has been carefully polished, and the authors’ substantial effort is evident. I recommend acceptance after minor revisions.

Reviewer #2: Thank you for the careful revisions. The manuscript has been clearly improved, the comments have been addressed, and I now find it suitable for publication.

**Do you want your identity to be public for this peer review?** For information about this choice, including consent withdrawal, please see our Privacy Policy

Reviewer #1: No

Reviewer #2: No

---

## [Author Response · Author response to Decision Letter 2]

9 Oct 2025

Dear Editors and Reviewers:

We sincerely thank the Editor for granting us the valuable opportunity to revise the manuscript for the second time. We also deeply appreciate the two reviewers for their continued time and effort: Reviewer 1 has provided constructive minor revision suggestions to refine the work and recommend acceptance after minor revisions, while Reviewer 2 has offered feedback recommending the direct acceptance of the manuscript.

Reviewer 1’s suggestions have effectively helped us identify targeted areas for optimization, which further enhances the rigor and completeness of the manuscript. We have carefully considered each point raised by Reviewer 1 and made focused, minor revisions to the manuscript accordingly. As for Reviewer 2’s recognition and acceptance recommendation, we are equally grateful for their positive evaluation of the work.

In response to the minor revision comments put forward by Reviewer 1 during this second round of revision, we hereby provide point-by-point replies.

We present the reviewers’ Comments in bold black text, our Responses in blue text, and the specific Revision Content in red text. (For the responses and corresponding revisions under each comment, we provide explanations in a point-by-point manner, ensuring that each response corresponds directly to the related revision.) Specifically, these are as follows:

Comment 1. The mechanism section is not yet sufficiently developed. I recommend strengthening it by drawing on the following five articles: (1)https://doi.org/10.1016/j.envres.2023.118074; (2)https://doi.org/10.1007/s11356-023-30307-z; (3)https://doi.org/10.1016/j.esr.2024.101590; (4)https://doi.org/10.1016/j.esr.2025.101837; (5)doi: 10.2166/wp.2025.355.

Response: Thank you for the valuable suggestion. Under your guidance, we carefully study these five articles of extremely high academic value and gain great insights into the application of quantile regression and spatial econometric methods. Therefore, to better refine the mechanism section of our paper, we carefully refer to these five articles to optimize the introduction, result presentation, and interpretation of the mechanism analysis.

Revision Content:

1)The Second paragraph of the “5.1 Channel Analysis”: Following prior researches (Ding et al., 2024; Chen et al., 2023; Imran et al., 2025; Imran et al., 2025; Ding et al.,2025), we employ quantile regression to better capture the asymmetric effects of digital transformation on corporate misconduct. The results reveal a “risk amplification effect”, whereby digital transformation intensifies misconduct risks among poorly governed firms. It indicates that digital transformation affects firms through specific mechanisms rather than uniformly increasing misconduct, thereby motivating a closer examination of the channels through which digital transformation shapes misconduct.

2)Footnote ⑥: The results show that at lower quantiles (10%-30%), the coefficients are positive but insignificant, indicating limited impact for firms with low misconduct propensity. From the 40th quantile onward, the coefficients become statistically significant (p < 0.05) and exhibit a monotonic upward trend, increasing from 0.86 at the 40th quantile to 1.45 at the 70th, and reaching a maximum of 1.97 at the 90th quantile (p < 0.1).

3)Lines 18-22 of the Fifth paragraph of the “6. Conclusion and Discussion”: In addition, the quantile regression results indicate that for insurers exposed to higher misconduct risks, digital transformation further amplifies propensity for misconduct, suggesting that it may create new opportunities for regulatory arbitrage. Future research could explore this issue in light of this asymmetric effect.

Comment 2. The introduction could more clearly articulate the research motivation—for example, why focus on China rather than the Middle East or South America, and why are samples from African regions important?

Response: Thank you for the valuable suggestion. Under your guidance, we further supplement our research motivation.

1)We have several reasons for focusing on China rather than other regions: First, in recent years, China’s financial industry generally undergoes digital transformation—especially after the implementation of the Broadband China Policy. At the same time, regulatory data show that as the digital transformation process advances, misconduct in China’s insurance industry is also on the rise. Second, China has unified compliance management standards, and data on insurance companies’ misconduct is accessible through relevant regulatory websites, which supports the conduct of our research.

2) Additionally, our research sample is from China rather than Africa, but the relevant description was not sufficiently clear; we have therefore supplemented and clarified this part as well.

Revision Content:

1) Lines 1-10 of the Second paragraph of the “1.Introduction”: “Corporate misconduct constitutes a unique operational risk, characterized by unethical business practices that break laws or harm stakeholders (Xia et al., 2023). Alarmingly, such misconducts exacerbate long-term inefficiencies, which in turn result in regulatory penalties, reputational damage, and market instability (Makau et al., 2021). The Chinese insurance market provides a representative context for analysis, as its regulatory framework is characterized by uniform rules and comprehensive information disclosure, offering consistent violation records and standardized penalty criteria. Consistently, regulatory data from China show an increase in insurers’ misconduct over the past decade—a trend that aligns with the rapid digital transformation of the insurance industry……”

2) Footnote ①: “Since the State Council launched the “Broadband China” plan in 2013, digital transformation in China has accelerated. Concurrently, data from the China Financial Regulatory Administration show that instances of insurers’ misconduct have increased alongside advancements in digital technology. Specifically, there has been a rapid increase in penalties for misconduct since 2017. In 2024, Chinese insurers received 2,762 misconduct penalties, with total fines amounting to 280 million yuan, representing an increase of 137.3% compared with 2017.”

3) First line of the fourth paragraph of the “1.introduction”: “Using data from Chinese insurers between 2010 and 2021, ……”.

Comment 3.The implications section could delve deeper into the study’s international relevance—for instance, what lessons might China’s experience offer for resource policies in developing countries across Asia and South America?

Response: Thank you for the valuable suggestion. Under your guidance, we have supplemented relevant content in the final chapter of the manuscript. Specifically, we delve into the transferable lessons of China’s experience for resource policies in developing countries, with a focused analysis of nations across Asia and South America. This supplementation strengthens the global perspective of our study.

Revision Content: The Third paragraph of the “6. Conclusion and Discussion”: “China's experience demonstrates that while digitalization often improves insurers’ profitability, it may also create incentives for misconduct that are frequently overlooked. This tension is especially salient for Asian and Latin American countries grappling with analogous challenges of limited insurance coverage and accelerated fintech proliferation. Therefore, understanding the misconduct risk during insurers’ digital transformation is crucial, as it directly affects industry integrity and the sustainability of digital innovation. Existing research largely emphasizes the efficiency and performance gains from digital transformation but pays limited attention to its unintended risks.”

Comment 4. The introduction should provide a more in-depth exposition of the research questions.

Response: Thank you for the valuable suggestion. Under your guidance:

1) We supplement the research background.

2) Prior to raising the research questions, we further supplement the rationale for doing so.

3) Clearly formulate the research question.

4) After putting forward the research questions, we supplement the value of addressing these questions.

Revision Content:

1) The same as the revision content of Comment 2.

2) Lines 10-12 of the Second paragraph of the “1.Introduction”: “This paradox between rapid digitalization and rising misconduct prompts critical questions about the unintended consequences of insurers’ digital transformation……”

3) Lines 12-15 of the Second paragraph of the “1.Introduction”: “……the following questions are raised: Are there risks of misconduct in the digital transformation of insurers? Why does the digital transformation of insurers present new challenges in managing misconduct while advancing the industry?......”

4) Lines 15-17 of the Second paragraph of the “1.Introduction”: “……Answering these questions offers valuable insights for regulators in emerging markets seeking to balance innovation and compliance.”

We would like to further thank all editors and reviewers. We have revised the manuscript in accordance with the comments, and hope it now meets the publication requirements. We kindly welcome any further feedback. Should there be any questions, please do not hesitate to contact us—we would be happy to provide explanations and make additional revisions to further improve the manuscript.

Kind regards,

Xiaoqing Guo, Ph.D Candidate

Corresponding Author

Central University of Finance and Economics

---

## [Editor Report · Decision Letter 2]

28 Oct 2025

Neglected Risks in the Chinese Insurers' Misconduct Caused by Digitalization

PONE-D-25-18596R2

Dear Dr. Guo,

We’re pleased to inform you that your manuscript has been judged scientifically suitable for publication and will be formally accepted for publication once it meets all outstanding technical requirements.

Kind regards,

Haijun Yang, Ph.D

Academic Editor

PLOS ONE
---

## [Editor Report · Acceptance letter]

PONE-D-25-18596R2

PLOS ONE

Dear Dr. Guo,

I'm pleased to inform you that your manuscript has been deemed suitable for publication in PLOS ONE. Congratulations! Your manuscript is now being handed over to our production team.

Kind regards,

on behalf of

Dr. Haijun Yang

Academic Editor

PLOS ONE